# Demulsification of Emulsion Using Heptanoic Acid during Aqueous Enzymatic Extraction and the Characterization of Peanut Oil and Proteins Extracted

**DOI:** 10.3390/foods12193523

**Published:** 2023-09-22

**Authors:** Tianci Li, Chenxian Yang, Kunlun Liu, Tingwei Zhu, Xiaojie Duan, Yandong Xu

**Affiliations:** College of Food Science and Engineering, Henan University of Technology, Zhengzhou 450001, China; 2021920112@stu.haut.edu.cn (T.L.); lkl@haut.edu.cn (K.L.); zhutingwei@haut.edu.cn (T.Z.); duanxj@haut.edu.cn (X.D.); xuyandong0929@126.com (Y.X.)

**Keywords:** aqueous enzymatic extraction, demulsification, heptanoic, oil characterization, functional properties

## Abstract

Peanut oil body emulsion occurs during the process of aqueous enzymatic extraction (AEE). The free oil is difficult to release and extract because its structure is stable and not easily destroyed. Demulsification can release free oil in an oil body emulsion, so various fatty acids were selected for the demulsification. Changes in the amount of heptanoic acid added, solid–liquid ratio, reaction temperature, and reaction time were adopted to investigate demulsification, and the technological conditions of demulsification were optimized. While the optimal conditions were the addition of 1.26% of heptanoic acid, solid–liquid ratio of 1:3.25, reaction temperature of 72.7 °C, and reaction time of 55 min, the maximum free oil yield was (95.84 ± 0.19)%. The analysis of the fatty acid composition and physicochemical characterization of peanut oils extracted using four methods were studied during the AEE process. Compared with the amount of oil extracted via other methods, the unsaturated fatty acids of oils extracted from demulsification with heptanoic acid contained 78.81%, which was significantly higher than the other three methods. The results of physicochemical characterization indicated that the oil obtained by demulsification with heptanoic acid had a higher quality. According to the analysis of the amino acid composition, the protein obtained using AEE was similar to that of commercial peanut protein powder (CPPP). However, the essential amino acid content of proteins extracted via AEE was significantly higher than that of CPPP. The capacity of water (oil) holding, emulsifying activity, and foaming properties of protein obtained via AEE were better than those for CPPP. Overall, heptanoic acid demulsification is a potential demulsification method, thus, this work provides a new idea for the industrial application of simultaneous separation of oil and proteins via AEE.

## 1. Introduction

Peanuts are one of the primary crops in the world and are an essential resource for oils and proteins [1]. In China, the planting area and the annual yield of peanuts show an increasing trend. The oil and protein content in peanuts is high, with the fat and protein contents reach 20–28% and 40–58%, respectively. Peanut oil is an edible oil that is rich in many kinds of unsaturated fatty acids. Moreover, peanut protein, as a high-quality edible plant protein resource, is rich in eight essential amino acids necessary for the human body and is extensively utilized in food because of its high utilization rate and useful functional properties [2]. The functional properties of proteins are essential for the sensory properties and texture of foods, especially the emulsifying properties, which play a crucial role in the formation and stability of emulsions [3,4]. Thus, the research on peanuts has a growing impact on human and economic development.

Aqueous enzymatic extraction (AEE) can realize simultaneous recovery of proteins and oils from oilseeds, taking water as a medium for extraction and separation with the assistance of enzymes [5,6]. Compared with high-temperature pressing or organic solvent extraction methods, AEE has many significant advantages, including product safety, being environmentally friendly, having milder reaction conditions, a lower energy consumption, and the possibility of capital investment which is more suitable for commercial processes [7,8,9]. Though AEE has many advantages, stable oil body emulsions are easily generate during the AEE process, while proteins and phospholipids as emulsifiers can form an interfacial membrane at the oil–water interface in the oil body emulsion, preventing aggregation of oil droplets and limiting oil extraction [10,11]. Therefore, the demulsification of oil body emulsions is essential for increasing oil production.

So far, many studies have reported physical, physicochemical, and enzymatic treatments as means of aqueous enzymatic extraction demulsification (AEED), which destroy interfacial membrane structure and promote oil droplet aggregation. After centrifugation, the oil is obtained [12,13,14]. Oil body emulsion via AEE is a thermodynamically unstable system. The higher temperature can alter the structure of proteins, which may negatively influence the stability of the oil body emulsion and be detrimental in increasing the yield of free oil [15]. Chabrand et al. [10] reported that freeze–thawing could promote oil coalescence since oil droplets crystallize and destroy the membranes of adjacent oil droplets under freezing conditions. Wu et al. [14] researched the effect of pH on the stability of soybean emulsion and found that the free oil content gradually increased with the pH decrease in soybean emulsion. When the pH of soybean emulsion was adjusted to the pI of soy protein, the soybean emulsion became destabilized and the free oil increased. In recent years, several studies have focused on adding enzymes to the emulsion to change the emulsion’s stability [16,17]. 

On the one hand, the interfacial proteins in the emulsion were hydrolyzed, their molecular size was reduced, and the adjacent oil droplet interfaces were decreased. On the other hand, the enzymatic reaction removed the high molecular weight polypeptides from the interfacial membrane, so the thickness of the interfacial membrane was reduced. This reaction contributed to oil droplets gathering and the obtention of more free oil [18]. However, physical and physicochemical treatments have the disadvantages of high energy consumption and low free oil content. Although demulsification via enzymes can increase the free oil content, enzymes are expensive, and the enzymatic reaction takes a long time. Therefore, searching for an effective, safe, and affordable approach to AEED is necessary. In our previous research, we found that adding a fatty acid could change the systemic environment of the emulsion and destroy the structure to achieve rapid demulsification [19]. In this paper, first, different fatty acids were screened based on the free oil yield (FOY). Second, to maximize free oil content, the effects of different demulsification parameters (the amount of heptanoic acid added, solid–liquid ratio, reaction temperature, and reaction time) were optimized using response surface methodology (RSD). Meanwhile, the fatty acid composition and physicochemical properties of oils obtained using different demulsification approaches were compared. In addition, the protein’s amino acid composition and functional properties through AEE were measured and compared with commercial peanut protein powder (CPPP). We expect this research to provide new ideas for oil body emulsion demulsification and a theoretical basis to promote the industrial production of AEEs.

## 2. Materials and Methods

### 2.1. Sample

Dehulled peanut seeds were obtained from the Zhengzhou Pengxun Seed Industry Co., Ltd. (Zhengzhou, China), called Yuanza-6 (YZ-6). Peanut seeds were roasted at 50 °C ovens to remove the red coat and stored at 4 °C. Viscozyme^®^ L was purchased from Novozymes Co., Ltd. (Shanghai, China). Several fatty acids were acquired from Macklin Co., Ltd (Shanghai, China), including valeric acid, hexanoic acid, heptanoic acid, octanoic acid, nonanoic acid, and decanoic acid.

### 2.2. Preparation of Oil Body Emulsion and Protein from AEE

The creation of an oil body emulsion via AEE was according to the method of Zhou et al. [20] with slight modifications. Figure 1 depicts the process flow for the oil body emulsion via AEE. Skinless peanuts (60 g) were soaked in fresh deionized water (seeds-to-water, 1:5) and kept at 4 °C for 12 h. Then, the mixture was pulverized under a multifunction food processor for 2 min according to the wet crushing process (JYL-C022E, Joyoung Co., Ltd., Jinan, China). Following this, enzyme (Viscozyme^®^ L) was introduced and stirred thermostatically for 2 h at 50 °C. After cooling to room temperature, the extracts were centrifuged at 5000 rpm for 15 min (DZ267-32C6; Anting Scientific Instrument Factory, Shanghai, China). The oil body emulsion was carefully removed, and the remaining aqueous and residual phases were centrifuged at 5000 rpm for another 10 min. Complete peanut oil body emulsions were eventually acquired, and the oil body emulsion obtained via AEE was stored at 4 °C for the next demulsification step. The proteins in the aqueous phase were isolated by alkali-dissolution and acid-precipitation, and their amino acid composition and functional properties were compared with CPPP.

### 2.3. Measurement of the Main Composition

The drying technique, the Kjeldahl, and the molybdenum blue colorimetric method [21,22] were applied to assess the moisture, protein, and phospholipid contents of peanuts and oil body emulsions, respectively. The total fat content of the peanuts was assessed by the Soxhlet extraction method according to Samira [23]. The oil content of oil body emulsions was determined using the chloroform–methanol method, with slight modifications, as described by Liu [21]. Specifically, three volumes of chloroform–methanol solution (2:1, *v*/*v*) were mixed with the oil body emulsion and stirred for 2 h. The filter residues were mixed with an equal volume of chloroform–methanol solution and stirred after the recovery of the extraction liquid. The aforementioned extraction process was carried out three times, then the extraction liquid was recycled. Rotary evaporation was used to remove the solvent. The residue was then dried for 10 h at 50 °C to achieve total fat. The neutral fat amount was calculated by deducting the phospholipid content from the total fat amount.

### 2.4. Oil Extracted by Different Demulsification Methods

#### 2.4.1. Heptanoic Demulsification

After weighing 5 g peanut oil body emulsion into a 50 mL beaker, a certain amount of deionized water at a solid–liquid ratio (1:4) was added. Then, the pH of the oil body emulsion was changed to 4.5 using the various fatty acids mentioned above, along with HCl serving as a control. The mixture was stirred in a water bath with a magnetic stirrer at a steady temperature (60 °C) for 40 min. After the reaction, the mixture was removed to centrifuge tubes (50 mL). Centrifugation at 5000 rpm for 10 min, resulted in an upper free oil layer, a stubborn emulsion layer, an aqueous layer, and precipitates. The upper free oil layer was removed and weighed. The FOY was calculated as in the following Equation (1):(1)$$\mathrm{The}\;\mathrm{FOY}\left(\%\right)=\frac{\mathrm{free}\;\mathrm{oil}\;\mathrm{content}\;\left(g\right)}{\mathrm{oil}\;\mathrm{body}\;\mathrm{emulsion}\left(g\right)\times \mathrm{oil}\;\mathrm{content}}\times 100$$

#### 2.4.2. Solvent Extraction (SE)

Oil body emulsion (50 g) by AEE was mixed with three volumes of chloroform–methanol solution (2:1, *v*/*v*), and stirred for 3 h. Then, the mixture was filtered and the supernatant collected. The extraction described above was repeated three times. The oils were collected and retained at 4 °C for the next character analysis after removing the solvent with a vacuum rotary evaporator (temperature 50 °C).

#### 2.4.3. Freeze–Thaw and Heat Treatment (FHD)

The freeze–thaw and heat treatment were used with the method reported by Li et al. [15]. Briefly, 30 g oil body emulsions extracted by AEE were frozen (−20 °C) for 24 h, then thawed in a water bath for 24 h at 50 °C. The defrosted emulsion was transferred into a centrifuge tube for centrifuging at 5000 rpm for 10 min. The oils were obtained and kept at 4 °C for the following character analysis.

#### 2.4.4. Isoelectric Point Demulsification (IPD)

After weighing 5 g peanut oil body emulsion into a 50 mL beaker, a certain amount of deionized water (solid–liquid ratio of 1:4) was added. Then, the pH of the oil body emulsion was adjusted to 4.5 using HCl and the mixture stirred at steady temperatures in a water bath at 60 °C for 40 min. After centrifugation, the free oil was obtained and stored at 4 °C.

### 2.5. Optimization of Demulsification Conditions with Heptanoic Acid

#### 2.5.1. Effects of Different Parameters on the FOY

Heptanoic acid demulsification parameters including the amount of heptanoic acid added (%), solid–liquid ratio (*w*/*v*), reaction temperature (°C), and reaction time (min) on the FOY were optimized by single-factor experimental design. The impact of the amount of heptanoic acid added on the FOY was explored under the following conditions: addition amount of fatty acid, 0.5, 1.0, 1.5, 2.0, and 2.5%; solid–liquid ratio at 1:4; reaction temperature at 70 °C; and reaction time for 40 min. Under the following factors, the impact of the solid–liquid ratio on FOY was examined: the amount of heptanoic acid added, 1.0%; solid–liquid ratio (1:2, 1:3, 1:4, 1:5, and 1:6); reaction temperature at 70 °C; reaction time for 40 min. The influence of reaction temperature on the FOY was investigated under the following conditions: the amount of heptanoic acid added, 1.0%; solid–liquid ratio of 1:4; reaction temperature at 40, 50, 60, 70, 80 °C; reaction time for 40 min. The consequence of reaction time on the FOY was studied under the following conditions: the amount of heptanoic acid added, 1.0%; solid–liquid ratio of 1:4; reaction temperature at 70 °C; reaction time for 20, 30, 40, 50, and 60 min.

#### 2.5.2. Experiment Design for Optimizing

Response Surface Methodology (RSM) was applied to evaluate the impact of various demulsification conditions on the FOY. Based on the above experimentations, the effects of four key parameters at three different variables were optimized by RSM according to the BBD. The range of the variables of various parameters involved was as follows: the amount of heptanoic acid added (0.5, 1.0, and 1.5%), solid–liquid ratio (1:3, 1:4, and 1:5 *w*/*v*), reaction temperature (60, 70, and 80 °C), and reaction time (40, 50, and 60 min). The FOY was taken as the response variable, and design expert software 11.0 version(Stat-Ease Inc., Minneapolis, MN, USA) was employed to assemble the experimental plan, analyze the regression of the data obtained, and evaluate the coefficient of the regression equation. In the experimental design process, the coded and actual levels of the independent variables were employed (Table 1), and the results summarized.

### 2.6. Determination of Composition of Fatty Acid

The fatty acid composition of peanut oils via various extract approaches was determined using gas chromatography (GC) by Gao et al. [24]. The samples (2–3 oil drops) were prepared by converting the oils extracted into fatty acid methyl esters (FAME). Then, the sample was injected into a GC-2014 instrument (Agilent 8990 N, Santa Clara, CA, USA), which had a BPX-70 capillary column (30.0 m × 250 μm × 0.25 μm) with flame ionization detector (FID). The sample was inserted in the following way: the inlet temperature was 250 °C, the detector temperature was 300 °C, with a split ratio of 1:50, and the nitrogen flow rate was 1.0 mL/min. The column temperature was maintained at 210 °C for 20 min after increasing by 2 °C/min from 170 °C. The NIST02 mass spectrometry library was used to match the identified chemical constituents with the mass spectrometry information and retention index of each chromatographic peak to determine the fatty acid composition of the peanut oils. The area normalization method was used for integral quantification.

### 2.7. Physicochemical Properties and Determination of Oil

The physicochemical properties of the oils were investigated under the optimized conditions of heptanoic demulsification and compared with other methods. The AOCS official techniques (Cd 3d-63, Cd 8b-90, Cd 1c-85, and Cd 3-25, respectively) were used to calculate the acid value, peroxide value, iodine value, and saponification value, while the detail on the determination is shown in Appendix A.

### 2.8. Amino Acid Composition Determination

The approach provided by Gao et al. [25] was used to analyze the amino acid composition and content with slight modifications. Briefly, the sample (20 mg) was put into a hydrolysis tube together with 6 mol/L HCl and phenol (3–4 drops). Then, the sample was hydrolyzed for 24 h at 110 °C. The hydrolysate tube was filled with nitrogen and put under vacuum. A sample diluent with a pH of 2.2 was used to dissolve the hydrolysate that resulted after drying, and an amino acid analyzer (S-433D, Sykam (Beijing, China) Scientific Instrument Co., Ltd., Beijing, China) was employed to determine the composition of the amino acids. The levels of 17 amino acids were examined separately. The amino acids were as follows: aspartate (Asp), threonine (Thr), serine (Ser), glutamate (Glu), glycine (Gly), alanine (Ala), cysteine (Cys), valine (Val), methionine (Met), isoleucine (Ile), leucine (Leu), tyrosine (Tyr), phenylalanine (Phe), histidine (His), lysine (Lys), arginine (Arg), and proline (Pro).

### 2.9. Determination of Functional Properties of Peanut Protein

#### 2.9.1. Solubility

An amount of 5 g protein was dissolved in deionized water (50 mL), and the pH adjusted to 7.0 using 1 M HCl or 1 M NaOH. After stirring at room temperature for 1 h and centrifuging at 5000 rpm for 20 min, the supernatant was collected. Using the Lowry method [26] the supernatant’s protein concentration was determined, utilizing the bovine serum albumin standard. The solubility of protein was determined as follows in Equation (2):(2)$$\mathrm{Solubility}\;(\%)=\frac{\mathrm{Protein}\;\mathrm{in}\;\mathrm{the}\;\mathrm{superstratum}}{\mathrm{Amount}\;\mathrm{of}\;\mathrm{all}\;\mathrm{protein}}\;\times 100$$

#### 2.9.2. Water-Holding Capacity (WHC) and Oil-Holding Capacity (OHC)

The WHC and OHC were determined with a slightly modified method reported by Jamdar [27]. The 0.5 g protein sample was placed in a centrifuge tube (10 mL), and deionized water (peanut oil, 5 mL) was added. After full oscillation and mixing, the sample was incubated in a water bath at room temperature for 30 min, centrifuged at 5000 r/min for 20 min, and the non-adsorbed water (peanut oil) on the upper supernatant was removed and weighed. The calculation of WHC and OHC was according to Equation (3) below:(3)$$\mathrm{WHC}\left(\mathrm{OHC}\right)=\frac{M_{2}-M_{1}}{M_{0}}$$

#### 2.9.3. Emulsion Activity Index (EAI) and Emulsion Stability Index (ESI)

The emulsification property of protein was carried out following Pearce et al. [28] with slight modifications. An amount of 5 mL of peanut oil and 15 mL of 0.2% protein solution (*w*/*v*, prepared with phosphate buffer at pH 7.0) were mixed with a high-speed mixer at 20,000 rpm for 1 min. Then, a 50 μL sample was taken from the bottom of the mixture and added to 5.0 mL 0.1% SDS solution. The absorption value was determined at 500 nm with the SDS solution as blank. EAI and ESI were calculated according to Equations (4) and (5) below:(4)$$\mathrm{EAI}(m^{2}/g)=\frac{2\times 2.303\times A_{0}\times N}{C\times \varphi \times 10,000}\times 100$$
(5)$$\mathrm{ESI}\left(\%\right)=\frac{A_{10}}{A_{0}}\times 100$$

A_0_, is the absorbance value at 0 min; A_10,_ the absorbance value at 10 min; N, dilution ratio; C, protein content in aqueous solution before creating an emulsion; and φ, the volume fraction of oil.

#### 2.9.4. Foaming Capacity (FC) and Foaming Stability (FS)

An amount of 0.5 g of protein was dissolved in 50 mL of phosphate buffer (10 mM/L, pH 7.0), stirred at room temperature for 30 min, and then homogenized at 10,000 r/min for 1 min. After immediately transferring all of the foam into a cylinder (50 mL), the volumes of foam V_1_ (0 min) and V_2_ were recorded after standing for 30 min. FC and FS were calculated following the formula below:(6)$$\mathrm{FC}\left(\%\right)=\frac{V_{1}}{20}\times 100$$
(7)$$\mathrm{FS}(\%)=\frac{V_{2}}{V_{1}}\times 100$$

### 2.10. Statistical Analysis

All experiments were carried out three times and the dates were presented as means ± SD. The mean values were compared by a one-way analysis of Variance (ANOVA) and the Waller–Duncan test in SPSS 26.0 software to evaluate the significance of the main effect. Significant differences were considered at *p*-value < 0.05. The data were plotted using Origin 2023.

## 3. Results and Discussion

### 3.1. The Main Composition of Peanut and Oil Body Emulsion

The results of the main composition of peanut and oil body emulsion are shown in Table 2. Peanut (YZ-6) had a fat content of (53.10 ± 0.24)%, protein content of (21.05 ± 0.18)% moisture content of (3.3 ± 0.06)%, ash content of (2.37 ± 0.05)%, and phospholipid content of (0.56 ± 0.04)%, respectively. Protein and oil are abundant in peanuts, which are suitable for extracting oil and protein. During the process of AEE, a small amount of protein and oil formed a stubborn emulsion. The peanut oil body emulsion contained (69.30 ± 1.07)% oil, (1.79 ± 0.24)% protein, (23.29 ± 0.64)% moisture, (0.2 ± 0.01)% ash, and (0.75 ± 0.03)% phospholipids.

### 3.2. Optimization of Demulsification of the Oil Body Emulsion

#### 3.2.1. Selection of Fatty Acid for Demulsification by AEE

Valeric acid, medium chain fatty acids (hexanoic acid, decanoic acid), lauric acid, and stearic acid were selected to demulsify the peanut oil body emulsion. The pH of the emulsion system was adjusted by different fatty acids to the pI of the oil body protein, and the FOYs of different fatty acids on the emulsion were studied, as shown in Figure 2. The pH of the emulsion system was adjusted by hydrochloric acid to the isoelectric point of oil body protein, and the FOY was (71.29 ± 0.81)%. The various fatty acids had different effects on the peanut oil body emulsion, among which the FOY of heptanoic acid and octanoic acid were the highest, which were (91.39 ± 0.53)% and (89.97 ± 0.30)%, respectively. Compared with hydrochloric acid, the FOYs of lauric acid and stearic acid were slightly lower than that of hydrochloric acid. The addition of other fatty acids could improve the demulsification rate of peanut oil body emulsion to different degrees, indicating that the addition of fatty acids could destroy the structure of the interface membrane of the oil body emulsion, reduce the stability of the emulsion, and release the oil [29]. Therefore, heptanoic acid was selected for subsequent tests.

#### 3.2.2. Effect of the Amount of Heptanoic Added on FOY

The effect of heptanoic acid addition on the FOY was investigated by varying from 0.5% to 2.5% (Figure 3a). With the increased amount of heptanoic acid added, the FOY showed an increase at first and then a decrease. When the concentration of heptanoic acid was 1%, the highest free oil content obtained was 93.08%. When the amount of heptanoic acid added was lower, it could not fully react with the oil body emulsion, so that it could not achieve demulsification. However, with a higher amount of heptanoic acid, the demulsification effect of heptanoic acid was inhibited, and the FOY was reduced.

#### 3.2.3. Effect of the Solid–Liquid Ratio on FOY

The effect of the solid–liquid ratio on FOY is shown in Figure 3b. With the increase of the solid–liquid ratio from 1:2 to 1:6, the FOY increased initially and subsequently declined. At the solid–liquid ratio of 1:4, the maximum free oil content was 94.28%. At a lower solid–liquid ratio (1:2 and 1:3), the emulsion was too viscous and difficult to react, so the FOY was decreased [20]. The ratio was greater than 1:4. On the one hand, the excessive dilution of the mixture could block the reaction of oil body emulsion with heptanoic acid; on the other hand, the oil droplets were too dispersed, and it was difficult to distinguish between oil droplets, resulting in a decrease in the FOY [30].

#### 3.2.4. Effect of Reaction Temperature on FOY

Temperature is an important factor in the demulsification of heptanoic acid. The effect of reaction temperature (40 °C to 80 °C) on the FOY was studied (Figure 3c). As the temperature increased (from 40 °C to 70 °C), the FOY increased significantly (*p* < 0.05). However, the FOY did not significantly increase when the temperature increased from 70 °C to 80 °C. The increase in temperature led to reduction of the viscosity of the oil body emulsion, which was more conducive to the diffusion of heptanoic acid in the oil body emulsion, and the Browne motion of oil droplets was increased [31]. Meanwhile, high temperatures expanded the interface protein structure of the oil body emulsion, exposed more binding sites, and improved the emulsion breaking rate of the oil body emulsion [32].

#### 3.2.5. Effect of Reaction Time on FOY

As shown in Figure 3d, it can be seen that the FOY rapidly increased with increasing time from 20 to 50 min and remained almost constant. At the optimum reaction time of 80 min, the free oil content achieved a maximum value of 94.66%. This could be attributed to the fact that the diffusion of heptanoic acid to the oil–water interface needed a certain amount of time to promote the completion of heptanoic acid and the oil body.

### 3.3. Response Surface Methodology (RSM) Analysis

RSM is an optimization method that graphically expresses the relationship between variables and response values to select the optimal conditions in the design of an experiment. Based on the above results, the amount of heptanoic acid added from 0.5% to 1.5%, the solid–liquid ratio from 1:3 to 1:5, reaction temperature from 60 to 80 °C, and reaction time range from 40 to 60 min were selected for BBD (shown in Table 1). The results of the RSM design and FOY are shown in Table 3. As shown, twenty-nine runs were carried out; taking FOY as the response and the actual levels of the independent variables used in the experimental design, the range of FOY was from 85.43 to 95.45%. Additionally, the significance of each model coefficient was assessed through the *t*-test and *p*-value. The relationships between the FOY (Y) and the amount of heptanoic acid added (X_1_), solid–liquid ratio (X_2_), reaction temperature (X_3_), and reaction time (X_4_) are represented by the response regression model equation as follows: Y = 94.09 + 1.58X_1_ − 1.57X_2_ + 0.81X_3_ + 0.75X_4_ − 3.16X_1_X_2_ + 0.31X_1_X_3_ + 0.05X_1_X_4_ − 0.29X_2_X_3_
 − 2.26X_2_X_4_ + 1.45X_3_X_4_ − 3.11X_1_^2^ − 2.84X_2_^2^ − 1.63X_3_^2^ − 2.40X_4_^2^

The ANOVA findings for the quadratic response surface model and multiple regression analysis were estimated using the relevant *F* and *p* values, and the results are displayed in Table 4. As described there, the *p*-value for the FOY model is less than 0.001 (*p* < 0.05 indicates significant), and “Lack of Fit” is 0.447 (>0.05), which demonstrates that the regression model is extremely significant and well fitted with the data. The established model’s determination coefficient was 0.9547, which indicates that the model could adequately predict the FOY response. Moreover, the correction determination coefficient predicted R^2^ of 0.7850 which is reasonably consistent with the adjusted R^2^ of 0.9095 (the difference was lower than 0.2), which reveals a high correlation between the observed data and the predicted data of the regression model [33]. The coefficient of variation is the percentage of the standard deviation to the mean value, and it is generally considered that less than 5% is considered reproducible. As depicted in Table 4, the CV for the FOY was 1.04%, implying that the FOY model is reproducible [34].

In the regression model for FOY, the variables X_1_, X_2_, X_3_, X_4_, interaction terms of X_1_X_2_, X_2_X_4_, X_3_X_4_, and all quadratic terms of X_1_^2^, X_2_^2^, X_3_^2^, X_4_^2^ were considered significant, while others were insignificant. In addition, the influence of factors on FOY could be determined according to the *F*-value. The greater the *F*-value, the greater is the influence of the factor on the test result. Therefore, from the *F*-value in Table 4, the order of influence for each variable is as follows: amount of heptanoic acid added > solid–liquid ratio > reaction temperature > reaction time.

### 3.4. Effect of Parameter Interactions on FOY

In order to intuitively visualize the mutual effects of two operational variables on FOY, a three-dimensional response surface curve was plotted. The trend of the response surface curve is used to demonstrate the interaction of various factors [35]. The trend of the response surface curve was steeper, and the interaction of two variables was more significant; otherwise, the interaction was less significant. The results showed the amount of heptanoic acid added and solid–liquid ratio had significant influences on FOY while reaction temperature and reaction time were not significant (Figure 4a–c). As described in Figure 4a–c, FOY increased with a fixed amount of heptanoic and decreased on increasing the solid–liquid ratio. As shown in Figure 4d and e, it reflects the interaction of the solid–liquid ratio with the two variables (reaction temperature and reaction time) on FOY. The FOY was the highest when the solid–liquid ratio was between 1:3 and 1:5. Moreover, the solid–liquid ratio and reaction temperature significantly influenced FOY. This was probably due to the high temperature helping to spread heptanoic acid to the oil-water interface, which could destroy the stability of the peanut oil body emulsion and increase FOY. As observed in Figure 4f, at a certain temperature, FOY increased with the initial reaction time, but the yield stability decreased with prolonged extraction. This might be due to the substrate having fully reacted at 55 min.

### 3.5. Verification of the Optimal Condition

A maximum FOY of 95.49% was predicted according to the regression equation model under the following optimum conditions: amount of heptanoic acid added was 1.266%, the solid–liquid ratio was 3.25, the reaction temperature was 72.74 °C, and the reaction time was 55.33 min. Considering the actual situation, verification tests in triplicate were carried out under the conditions of 1.26% heptanoic acid addition, solid–liquid ratio 1:3.25, reaction temperature 72.7 °C, and reaction time 55 min. The final actual FOY was (95.84 ± 0.19)%, which was close to the predicted value, indicating that the regression model obtained was accurate and reliable in predicting the demulsification rate on the oil body emulsion. Compared to other demulsification procedures, heptanoic acid demulsification gives a greater free oil content, requires a shorter demulsification time, is inexpensive, and suitable for industrial applications.

### 3.6. Oil Character Analsisy

#### 3.6.1. Fatty Acid Composition Analysis of Different Oils

The fatty acid compositions of peanut oils obtained via different demulsifications were analyzed. As shown in Table 5, eight main compositions, two polyunsaturated fatty acids (PUFA), one mono-unsaturated fatty acid (MUFA), and five saturated fatty acids (SAFA) were determined. These results were consistent with those reported by Zhao et al. [36]. Even though the types of fatty acids did not give a difference, their contents showed significant differences (*p* < 0.05) with different demulsification. Peanut oil unsaturated fatty acid content ranged from 78.26% to 78.81%. Compared with others, the unsaturated fatty acid content of peanut oil obtained by heptanoic demulsification was the highest. This might be due to the high contents of oleic acid and linoleic acid in peanut oil, which were 38.02% and 39.97%, respectively. Oleic acid is a mono-unsaturated fatty acid that can effectively lower the body’s low-density lipoprotein cholesterol (LDL) levels and maintain high-density lipoprotein cholesterol (HDL) levels, essential for maintaining cardiovascular health [37,38]. Linoleic acid is a kind of unsaturated fatty acid that cannot be synthesized by the body but is indispensable for maintaining the body’s life activities. Peanut oil is an important source of linoleic acid, which is needed by the body. The O/L value represents the ratio of oleic acid to linoleic acid, which is connected to the storage period of peanut oil [39]. The higher the O/L value, the less is the rancidity of peanut oil and the longer the storage period. On the contrary, peanut oil is prone to oxidation and spoilage [40]. Gao et al. [25] compared the fatty acid composition by cold-pressed and AEE and found there was no significant difference between the two methods. Still, the contents of the unsaturated fatty acids were consistent with our results. All in all, different demulsification methods significantly affect peanut oil fatty acid content. Peanut oil extracted from demulsification with heptanoic acid had higher unsaturated fatty acid content, a longer storage time, and a higher fat quality.

#### 3.6.2. Physicochemical Character Analysis of Different Oils

Under optimal heptanoic demulsification conditions, the physicochemical properties of the oil were compared with those of the oil demulsified by SE, FTHD, and IPD. As shown in Table 6, the acid, peroxide, saponification, and iodine values of oil extracted via the four methods had significant differences (*p* < 0.05) according to ANOVA analysis. The acid value and peroxide value are essential indices to assess oil character. The acid value represents the content of free oil fatty acid, while the peroxide value indicates the degree of oxidation of the oil. Notably, the acid value of oil of HD was significantly higher than SE, FTHD, and IPD, but the peroxide value was lower than the other three methods. The difference in acid value might be because part of the triglycerides decompose into free fatty acids in the process of heptanoic acid demulsification, while the introduction of foreign fatty acids may be another reason. The saponification value of oil extracted by different methods ranged from 175.84 ± 0.11 to 189.2 ± 0.07 KOH/g. The oils obtained via HD and FTHD had the highest saponification values, which illustrated that the oils contained more low molecular weight and short-chain fatty acids. The iodine value assesses the degree of unsaturated oil. The iodine value (109.33 ± 0.14 g I_2_/100 g) of oil obtained via HD was higher than those of oil extracted by SE (106.81 ± 1.21 g I_2_/100 g), FTHD (108.39 ± 0.74 g I_2_/100 g), and IPD (106.57 ± 1.1 g I_2_/100 g). This result demonstrated that the oil had higher unsaturated fatty acid, which was consistent with the fatty acid composition. The above results showed that the demulsification with heptanoic acid is an effective and feasible method, and the oil quality obtained is higher than that of the traditional method. Thus, the demulsification with heptanoic acid has a good development prospect.

### 3.7. Amino Acid Composition Determination

The process of plant oil during AEE allowed the simultaneous separation of peanut proteins in the aqueous phase and improved the resource utilization of peanuts [41,42]. The amino acid composition of peanut proteins extracted via AEE was compared with CPPP and is shown in Table 7. There was a significant difference (*p* < 0.05) in the composition of peanut protein extracted via AEE and CPPP. As shown in Table 7, the hydrophobic amino acid content of CPPP was higher than that of AEEP. However, the essential amino acid content of proteins extracted via AEE was significantly higher than that of CPPP, which showed that the proteins extracted via AEE had high nutritional value and were more suitable for human digestion and absorption.

### 3.8. Functional Property of Proteins

In order to further evaluate the quality of protein extracted via AEE, the functional properties of the two proteins were compared, as shown in Table 8. The results showed that the solubility of AEEP was lower than that of CPPP, which might be due to the fact that Viscozyme^®^ L used in aqueous enzymatic extraction acted on the cell well but had no negative effect on the molecular weight and structure of the protein, resulting in poor solubility [43]. However, the emulsifying activity and foaming properties of AEEP were significantly higher than CPPP (*p* < 0.05), and the emulsifying stability and foaming stability of AEEP were higher than CPPP. The capacity of the protein to bind to water is usually expressed by water holding. As shown in Table 8, the water holding capacity of AEEP (1.89 ± 0.11 g/g) is better than that of CPPP (1.41 ± 0.03 g/g) because CPPP was denatured, and its structure changed due to degreasing with organic solvent during extraction. The ability of a protein to hold oil might affect its ability to emulsify. The stronger the oil holding capacity of protein, the more likely it can effectively enhance the absorption and storage capacity of food for oil and reduce the loss of oil in the production process [44,45]. The oil-holding ability of AEEP was 2.43 ± 0.01 g/g, which was higher than that of CPPP (2.25 ± 0.05 g/g). The results were similar to those reported by Liu et al. [21], but the functional properties of AEEP in this paper were better. Overall, the results above showed that the peanut protein extracted via AEE had good properties and could effectively retain the structure and function of the protein, thus being more suitable for application in the food industry.

## 4. Conclusions

In this study, demulsification variables of heptanoic were optimized to improve the FOY. The optimized conditions were obtained as follows: amount of heptanoic acid added of 1.26%, solid–liquid ratio of 1:3.25, reaction temperature of 72.7 °C, and reaction time of 55 min. The FOY can be up to 95.84 ± 0.19% under optimized conditions. The oil obtained via heptanoic demulsification has higher unsaturated fatty acids that can benefit human health. According to the analysis of the physicochemical properties, the peanut oil obtained via heptanoic acid demulsification had better antioxidant qualities and a longer shelf life than other methods. The amino acid analysis showed that AEEP had a higher content of essential amino acids and was more nutritious. Compared with CPPP, the solubility of AEEP was lower; however, the oil-holding (water-holding), emulsifying properties, and foaming properties were better than for CPPP. In summary, AEE has a broad industrial prospect in the simultaneous separation of protein and oil, while heptanoic acid demulsification provides a new strategic approach for the demulsification of the oil body extracted via AEE.

## Figures and Tables

**Figure 1 foods-12-03523-f001:**
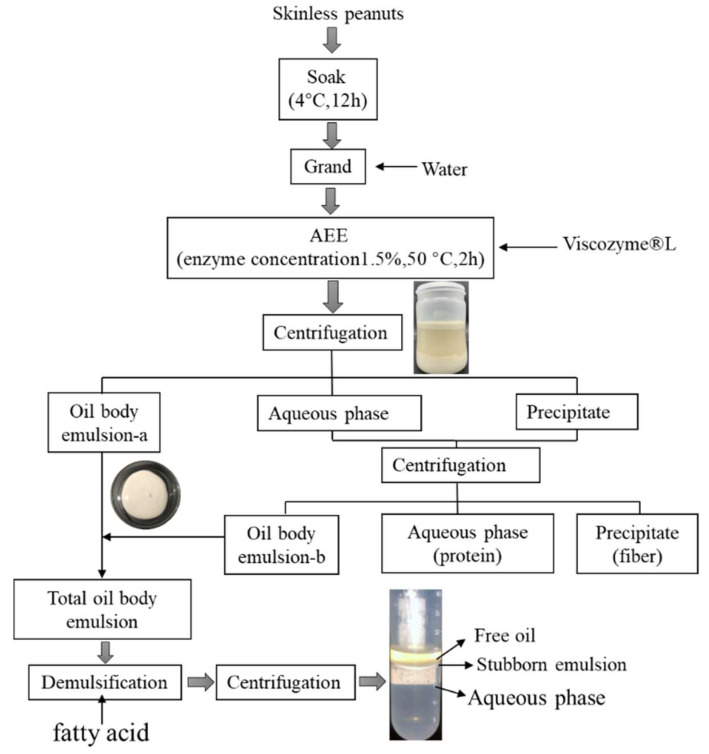
Process flow sheet for AEE of peanut oil body emulsion and protein.

**Figure 2 foods-12-03523-f002:**
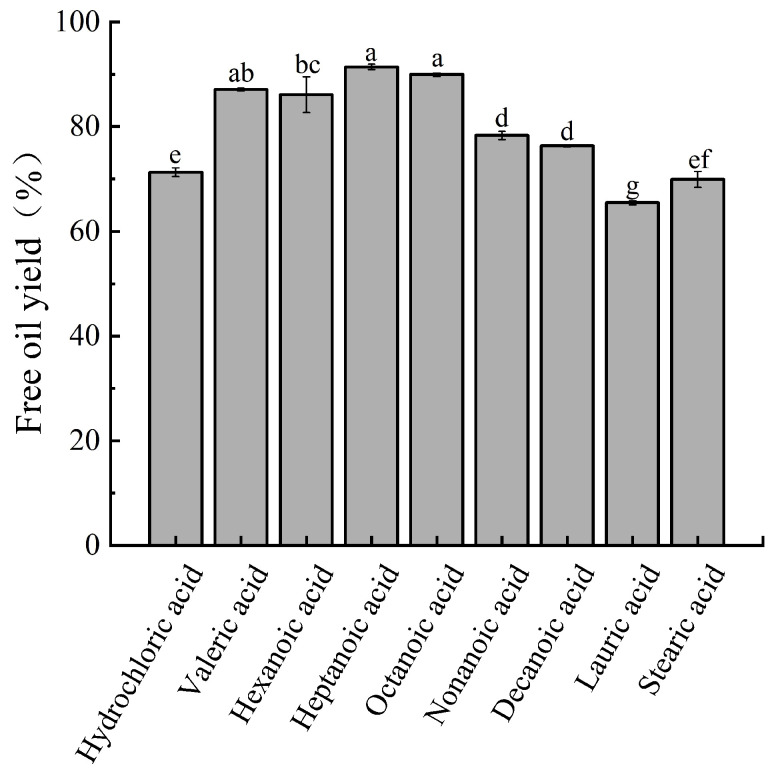
The influence of different fatty acids on FOY. A significant difference between samples is indicated by different lowercase letters (*p* < 0.05).

**Figure 3 foods-12-03523-f003:**
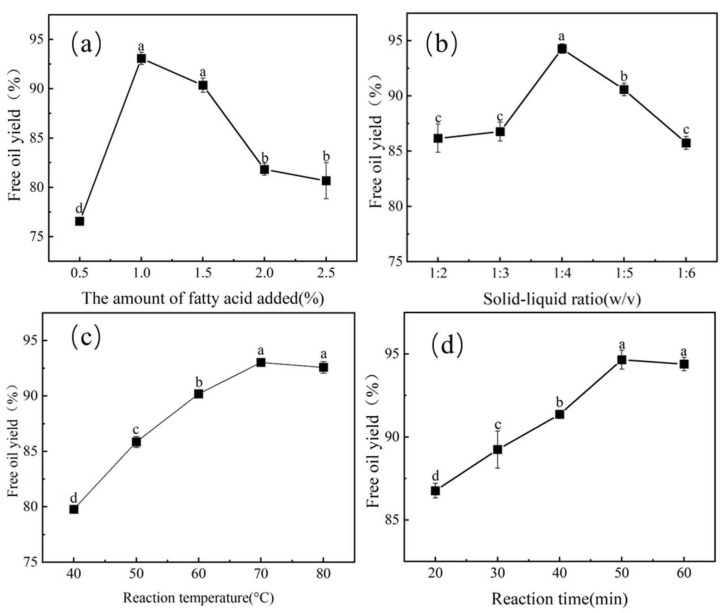
Effect of different parameters on FOY. (**a**) The amount of heptanoic added, (**b**) solid–liquid ratio, (**c**) reaction temperature, (**d**) reaction time. A significant difference between samples is indicated by different lowercase letters (*p* < 0.05).

**Figure 4 foods-12-03523-f004:**
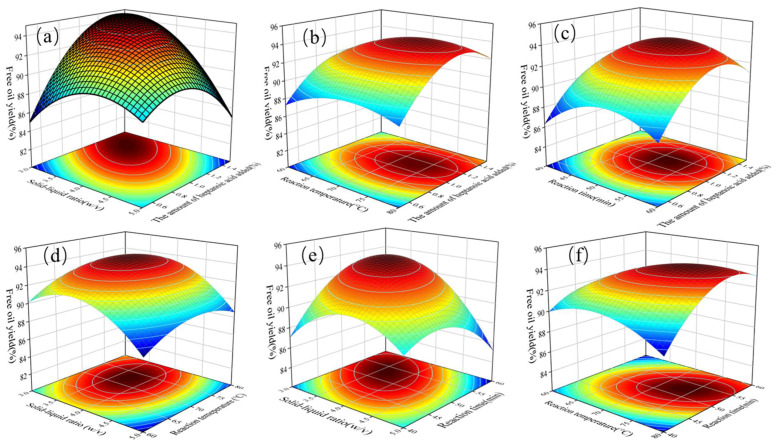
The response surface plot for the mutual effects of two different variables on FOY. (**a**) The addition amount of heptanoic and solid–liquid ratio, (**b**) the amount of heptanoic acid added and reaction temperature, (**c**) the amount of heptanoic acid added and reaction time, (**d**) solid–liquid ratio and reaction temperature, (**e**) solid–liquid ratio and reaction time, (**f**) reaction temperature and reaction time.

**Table 1 foods-12-03523-t001:** Independent variables and their levels used in BBD.

Factor Levels	Independent Variables			
	X_1_	X_2_	X_3_	X_4_
	amount of fatty acid added (%)	solid–liquid ratio (*w*/*v*)	reaction temperature (°C)	reaction time (min)
−1	0.5	1:3	60	40
0	1.0	1:4	70	50
1	1.5	1:5	80	60

**Table 2 foods-12-03523-t002:** Main components analysis of peanut and oil body emulsion.

	Oil (%)	Protein (%)	Water (%)	Ash (%)	Phospholipid (%)
Peanut	53.10 ± 0.24	21.05 ± 0.18	3.30 ± 0.06	2.37 ± 0.05	0.56 ± 0.04
oil body emulsion	69.30 ± 1.07	1.79 ± 0.24	23.29 ± 0.64	0.2 ± 0.01	0.75 ± 0.03

**Table 3 foods-12-03523-t003:** Response Surface Methodology and the corresponding experimental results obtained by BBD.

Run	X_1_ (%)	X_2_ (*w*/*v*)	X_3_ (°C)	X_4_ (min)	Y (%)
1	0	0	−1	−1	90.56
2	0	0	−1	1	88.97
3	−1	0	0	1	86.78
4	−1	1	0	0	89.45
5	1	0	0	−1	89.86
6	0	0	0	0	94.06
7	−1	0	−1	0	86.9
8	1	0	−1	0	90.08
9	0	1	0	1	86.1
10	0	0	1	1	93.51
11	0	−1	0	1	93.8
12	0	0	1	−1	89.32
13	0	−1	−1	0	89.97
14	−1	−1	0	0	85.59
15	1	0	0	1	90.29
16	−1	0	1	0	87.38
17	1	0	1	0	91.78
18	0	1	0	−1	87.76
19	0	0	0	0	94.19
20	0	1	−1	0	86.77
21	1	1	0	0	85.43
22	0	1	1	0	88.29
23	1	−1	0	0	94.21
24	0	−1	1	0	92.64
25	0	0	0	0	93.56
26	0	0	0	0	93.2
27	−1	0	0	−1	86.56
28	0	0	0	0	95.45
29	0	−1	0	−1	86.43

**Table 4 foods-12-03523-t004:** Analysis of variance for the regression model of FOY.

Source	Sum of Squares	Degree of Freedom	Mean of Square	*F* Value	*p* Value	Significant
Model	256.65	14	18.33	21.09	<0.0001	**
X_1_	30.05	1	30.05	34.57	<0.0001	**
X_2_	29.58	1	29.58	34.03	<0.0001	**
X_3_	7.79	1	7.79	8.96	0.0097	*
X_4_	6.69	1	6.69	7.70	0.0149	*
X_1_X_2_	39.94	1	39.94	45.95	<0.0001	**
X_1_X_3_	0.3721	1	0.3721	0.4280	0.5236	NS
X_1_X_4_	0.0110	1	0.0110	0.0127	0.9119	NS
X_2_X_3_	0.3306	1	0.3306	0.3803	0.5473	NS
X_2_X_4_	20.39	1	20.39	23.45	0.0003	*
X_3_X_4_	8.35	1	8.35	9.61	0.0078	*
X_1_^2^	62.68	1	62.68	72.10	<0.0001	**
X_2_^2^	52.40	1	52.40	60.28	<0.0001	**
X_3_^2^	17.15	1	17.15	19.73	0.0006	*
X_4_^2^	37.51	1	37.51	43.15	<0.0001	**
Residual	12.17	14	0.8693			
Lack of fit	9.24	10	0.9237	1.26	0.4447	NS
Pure error	2.93	4	0.7334			
Cor total	268.82	28				
R^2^	0.9547					
Adj.R^2^	0.9095					
Pre.R^2^	0.7850		C.V.	1.04		

Note: * and ** represent *p* < 0.05 and *p* < 0.01, respectively, and NS represent not significant.

**Table 5 foods-12-03523-t005:** Fatty acid composition of peanut oils obtained by different demulsification methods.

Fatty Acids	SE	HD	FTHD	IPD
Palmitic acid (16:0)	12.32 ± 0.01 ^a^	11.97 ± 0.02 ^a^	11.95 ± 0.04 ^b^	12.00 ± 0.02 ^a^
Stearic acid (18:0)	4.01 ± 0.02 ^a^	3.95 ± 0.00 ^ab^	3.95 ± 0.01 ^b^	4.00 ± 0.04 ^ab^
Oleic acid (18:1)	37.90 ± 0.02 ^ab^	38.02 ± 0.04 ^a^	37.65 ± 0.13 ^c^	37.70 ± 0.1 ^bc^
Linoleic acid (18:2)	39.53 ± 0.03 ^c^	39.97 ± 0.01 ^b^	40.27 ± 0.07 ^a^	40.01 ± 0.01 ^b^
Arachidic acid (20:0)	1.56 ± 0.00 ^a^	1.56 ± 0.01 ^a^	1.55 ± 0.01 ^a^	1.57 ± 0.01 ^a^
Arachidonic acid (20:1)	0.83 ± 0.01 ^ab^	0.82 ± 0.02 ^b^	0.82 ± 0.00 ^ab^	0.86 ± 0.00 ^a^
Behenic acid (22:0)	2.48 ± 0.02 ^ab^	2.43 ± 0.01 ^b^	2.46 ± 0.04 ^ab^	2.52 ± 0.02 ^a^
Tetracosanoic acid (24:0)	1.37 ± 0.02 ^a^	1.28 ± 0.05 ^a^	1.35 ± 0.04 ^a^	1.34 ± 0.04 ^a^
SFA	21.74	21.19	21.25	21.43
UFA	78.26	78.81	78.75	78.56
O/L	0.93	0.95	0.96	0.94

Note: SE, HD, FTHD, IPD represent the oils obtained by different methods of solvent extraction, heptanoic acid demulsification, freeze–thaw, heat treatment, and isoelectric point demulsification. A significant difference between samples is indicated by different lowercase letters (*p* < 0.05).

**Table 6 foods-12-03523-t006:** Physicochemical properties of peanut oils obtained by different demulsification methods.

Physicochemical Properties	SE	HD	FTHD	IPD
acid value (mg KOH/g)	0.43 ± 0.02 ^c^	0.63 ± 0.02 ^a^	0.39 ± 0.00 ^c^	0.55 ± 0.02 ^b^
peroxide value (g/100 g)	0.08 ± 0.00 ^c^	0.07 ± 0.00 ^d^	0.09 ± 0.00 ^b^	0.10 ± 0.00 ^a^
saponification value (KOH/g)	175.84 ± 0.11 ^c^	189.2 ± 0.07 ^a^	187.09 ± 1.32 ^a^	184.11 ± 0.67 ^b^
iodine value (g I_2_/100 g)	106.81 ± 1.21 ^ab^	109.33 ± 0.14 ^a^	108.39 ± 0.74 ^ab^	106.57 ± 1.1 ^b^

Note: SE, HD, FTHD, and IPD represent the oils obtained by different methods of solvent extraction, heptanoic acid demulsification, freeze–thaw, heat treatment, and isoelectric point demulsification. A significant difference between samples is indicated by different lowercase letters (*p* < 0.05).

**Table 7 foods-12-03523-t007:** Amino acid composition of different proteins.

Amino Acid	AEEP	CPPP
Aspartic acid (Asp)	11.94 ± 0.01 ^a^	11.61 ± 0.01 ^b^
* Threonine (Thr)	3.23 ± 0.06 ^a^	3.31 ± 0.01 ^a^
Serine (Ser)	5.27 ± 0.01 ^a^	5.27 ± 0.01 ^a^
Glutamic acid (Glu)	18.06 ± 0.01 ^a^	17.86 ± 0.01 ^a^
Glycine (Gly)	3.89 ± 0.01 ^a^	4.145 ± 0.05 ^b^
Alanine (Ala)	3.06 ± 0.01 ^a^	3.11 ± 0.01 ^a^
Cystine (Cys)	2.28 ± 0.06 ^a^	2.12 ± 0.01 ^a^
* Valine (Val)	3.66 ± 0.01 ^a^	3.79 ± 0.01 ^b^
* Methionine (Met)	2.755 ± 0.01 ^a^	2.7 ± 0.01 ^b^
* Isoleucine (Ile)	4.1 ± 0.01 ^a^	3.97 ± 0.01 ^b^
* Leucine (Leu)	6.83 ± 0.01 ^a^	6.62 ± 0.03 ^b^
Tyrosine (Tyr)	5.41 ± 0.01 ^a^	4.79 ± 0.03 ^a^
* Phenylalanine (Phe)	6 ± 0.01 ^a^	5.62 ± 0.01 ^a^
Histidine (His)	3.25 ± 0.01 ^a^	3.17 ± 0.01 ^b^
Lysine (Lys)	4.22 ± 0.01 ^a^	4.54 ± 0.01 ^b^
Arginine (Arg)	10.87 ± 0.04 ^a^	11.265 ± 0.04 ^b^
Proline (Pro)	5.16 ± 0.01 ^a^	6.13 ± 0.00 ^a^
Essential AA	30.80 ± 0.02 ^a^	30.54 ± 0.06 ^b^
Hydrophobic AA	35.46 ± 0.10 ^b^	36.08 ± 0.01 ^a^

Note: AEEP stands for proteins by aqueous enzymatic extraction, and CPPPP stands for commercial peanut protein powder. A significant difference between samples is indicated by different lowercase letters (*p* < 0.05). *, stands for essential amino acid.

**Table 8 foods-12-03523-t008:** Functional properties of different peanut proteins.

Functional Property	Solubility (%)	Water Holding (g/g)	Oil Holding (g/g)	Emulsifying Activity (m^2^/g)	Emulsifying Stability (%)	Foaming Property (%)	Foaming Stability (%)
AEEP	11.95 ± 0.16 ^b^	1.89 ± 0.11 ^a^	2.43 ± 0.01 ^a^	45.71 ± 0.12 ^a^	58.44 ± 1.79 ^a^	20.70 ± 0.14 ^a^	52.66 ± 0.32 ^a^
CPPP	12.96 ± 0.35 ^a^	1.41 ± 0.03 ^b^	2.25 ± 0.05 ^b^	39.70 ± 0.98 ^b^	53.29 ± 1.55 ^a^	19.60 ± 0.28 ^b^	51.55 ± 1.46 ^a^

Note: AEEP stands for proteins by aqueous enzymatic extraction, and CPPPP stands for commercial peanut protein powder. A significant difference between samples is indicated by different lowercase letters (*p* < 0.05).

## Data Availability

The data are not publicly available due to confidentiality requirements for dissertations.

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
