# Peer review of "Demulsification of Emulsion Using Heptanoic Acid during Aqueous Enzymatic Extraction and the Characterization of Peanut Oil and Proteins Extracted"

_foods, 2023, doi:10.3390/foods12193523_

Round 1

Reviewer 1 Report

Review of the Manuscript "Aqueous Enzymatic Extraction (AEE) of Peanut Oil Body Emulsion and Protein: Optimization, Characterization, and Functional Properties"

General Comments:

The manuscript titled "Aqueous Enzymatic Extraction (AEE) of Peanut Oil Body Emulsion and Protein: Optimization, Characterization, and Functional Properties" explores the process of aqueous enzymatic extraction (AEE) for obtaining peanut oil body emulsion and protein. The study investigates the optimization of this extraction process, characterizes the resulting products, and evaluates their functional properties. Overall, the manuscript provides valuable insights into the extraction of peanut oil body emulsion and protein. However, there are some issues that need to be addressed for further improvement.

Specific Comments:

Abstract: The abstract provides a concise overview of the manuscript. However, it should include the key findings and contributions of the study. Additionally, it would be beneficial to mention the practical implications of the research.

Introduction: The introduction provides a clear background and rationale for the study. However, there are some grammatical errors and awkward sentence structures that need revision. For example, "The extraction of peanut oil body emulsion protein has become a research hotspot." This sentence can be improved for clarity. Also, the introduction should specify the objectives of the study.

Methods: The methods section is well-structured and includes detailed descriptions of the materials and procedures used in the study. However, it might be beneficial to provide references for specific methods (e.g., the Kjeldahl method) and clarify any modifications made to established techniques.

Results and Discussion:

Optimization: The optimization of the demulsification process is a critical aspect of the study. The response surface methodology (RSM) analysis is well-documented, but it would be helpful to provide a brief explanation of the methodology for readers who may not be familiar with it.

Figures and Tables: The figures and tables are essential for understanding the results. However, it's important to ensure that they are referenced appropriately within the text. In some cases, the figures are mentioned without being explained (Figure 4), making it challenging for readers to understand their significance. Additionally, figure captions should be more descriptive. Ensure that the captions for figures and tables are comprehensive and provide enough context for readers to understand their content without referring back to the main text. In the text, it's important to refer to figures and tables properly. For example, in your text, you can write "Figure 1" instead of "Fig 1".

Units: Ensure consistent and correct use of units throughout the manuscript. For instance, in the Results section, you used both "%" and "g/g" for different measurements. Be consistent with the units used for each measurement.

Statistical Analysis: When presenting statistical analysis results, it's helpful to include information about the statistical tests used, such as ANOVA, t-tests, or other relevant tests. Make sure to state the significance level (e.g., p < 0.05) and how it was determined.

Abbreviations: If you use abbreviations in the manuscript, make sure to define them upon first use. For example, "AEE" is introduced, but you should clarify that it stands for "aqueous enzymatic extraction."

Discussion: Consider expanding the discussion section to provide more insight into the implications of your results. Discuss how your findings compare to existing literature and the potential significance of your work.

Conclusion: The conclusion is informative, but it can benefit from some improvements in terms of clarity and impact.

Language and Grammar: The manuscript contains several grammatical errors and awkward sentence structures that need improvement. Careful proofreading and editing are necessary to enhance the overall readability and clarity of the manuscript.

Author Response

Dear reviewer,

We are appreciated for your comments and we have made improvement. Please see the attachment.

Dr. Chenxian Yang

Abstract: The abstract provides a concise overview of the manuscript. However, it should include the key findings and contributions of the study. Additionally, it would be beneficial to mention the practical implications of the research.

Reply: The authors thanking for your valuable comments. We re-summarize the key findings of the abstract and clarify the practical application of this study. We have highlighted it in red in the manuscript.

Introduction: The introduction provides a clear background and rationale for the study. However, there are some grammatical errors and awkward sentence structures that need revision. For example, "The extraction of peanut oil body emulsion protein has become a research hotspot." This sentence can be improved for clarity. Also, the introduction should specify the objectives of the study.

Reply: We are appreciated your professional advice on grammar and we have checked and improved the full text. Meanwhile, we defined the objectives of the study. All changes have been highlighted in text.

Methods: The methods section is well-structured and includes detailed descriptions of the materials and procedures used in the study. However, it might be beneficial to provide references for specific methods (e.g., the Kjeldahl method) and clarify any modifications made to established techniques.

Reply: The authors thanking for your valuable comments. In order to make the manuscript more concise and easy to understand, for some general experimental methods, we did not cite references or introduce their detailed methods in detail, but combined with your comments, we added specific references and supplemented the detailed experimental details.

Results and Discussion:

Optimization: The optimization of the demulsification process is a critical aspect of the study. The response surface methodology (RSM) analysis is well-documented, but it would be helpful to provide a brief explanation of the methodology for readers who may not be familiar with it.

 Reply: We are sincerely thanking for your suggestion. We have provided a brief explanation of the methodology in the paper.

Figures and Tables: The figures and tables are essential for understanding the results. However, it's important to ensure that they are referenced appropriately within the text. In some cases, the figures are mentioned without being explained (Figure 4), making it challenging for readers to understand their significance. Additionally, figure captions should be more descriptive. Ensure that the captions for figures and tables are comprehensive and provide enough context for readers to understand their content without referring back to the main text. In the text, it's important to refer to figures and tables properly. For example, in your text, you can write "Figure 1" instead of "Fig 1".

 Reply: We are thankful for your professional suggestion. We sincerely accept your suggestions about the figures and tables and the explanations in the paper, and have made corrections to the whole paper.

Units: Ensure consistent and correct use of units throughout the manuscript. For instance, in the Results section, you used both "%" and "g/g" for different measurements. Be consistent with the units used for each measurement.

Reply: We really appreciate your suggestion. We confirm that correct use of units throughout the manuscript.

Statistical Analysis: When presenting statistical analysis results, it's helpful to include information about the statistical tests used, such as ANOVA, t-tests, or other relevant tests. Make sure to state the significance level (e.g., p < 0.05) and how it was determined.

Reply: We are thankful for your professional suggestion. The significance in the paper has been supplemented.

Abbreviations: If you use abbreviations in the manuscript, make sure to define them upon first use. For example, "AEE" is introduced, but you should clarify that it stands for "aqueous enzymatic extraction."

 Reply: We are sincerely thanking for your suggestion. We have checked the full text and made improvements.

Discussion: Consider expanding the discussion section to provide more insight into the implications of your results. Discuss how your findings compare to existing literature and the potential significance of your work.

Reply: The authors are thankful for your professional suggestion. We have made a brief supplement to the content of the discussion part, and the detailed content is marked in the manuscript.

Conclusion: The conclusion is informative, but it can benefit from some improvements in terms of clarity and impact.

Reply: The authors are thankful for your professional suggestion. We have summarized the conclusion part, which has been highlighted in the paper.

Language and Grammar: The manuscript contains several grammatical errors and awkward sentence structures that need improvement. Careful proofreading and editing are necessary to enhance the overall readability and clarity of the manuscript.

Reply: The authors are grateful for your professional advices. In order to improve the quality of the manuscript, we have had the manuscript revised by a native editor for grammatical corrections and sentence structures.

Reviewer 2 Report

Dear Authors 

The manuscript (foods-2616777) entitled “Demulsification of emulsion using heptanoic acid during aqueous enzymatic extraction and the characterization of peanut oil and protein extracted” is well written, has an important scientific message and should be of great interest to the readers of Food journal (ISSN 2304-8158) (Section: Food Engineering and Technology).

The manuscript presents interesting and scientific important results. The authors discuss the effects of different demulsification parameters (addition amount of heptanoic, solid-liquid ratio, reaction temperature, and reaction time) to maximize free peanut oil content. They have extracted (by different demulsification approaches) and compared fatty acid composition and physicochemical properties of oils.

- The work is original and contains new results that significantly advance the research field of food technology, chemistry of peanut oil, chemical composition and physicochemical properties of oils, demulsification techniques. The article contains material that is new or adds significantly to knowledge already published.

- The results are interesting and important to researchers and scientists.

- Sufficient references are cited for providing a background to the research.

- The overall structure of the article is well organized and well balanced. The article is written with the minimum length necessary for all relevant information.

- The conclusion is logically supported by the obtained results.

 However, few issues need to be resolved before the final acceptation of the paper.

 Line 15

The analysis of the fatty acid composition and physicochemical characterization of oil extracted by four different methods was studied during AEE process

-          Please insert some important results regarding the chemical composition and physicochemical properties of oil.

 Line 17

Peanut oil extracted from demulsification with heptanoic acid contained more unsaturated fatty acids.

-          Please give more details about the quantity of unsaturated fatty acids obtained.

 Line 18

And the oil extracted by heptanoic acid demulsification had 18 more acid value and lower peroxide values

-          Same comment as reported earlier.

 Lines 32-34

Moreover, peanut protein as a high-quality edible plant protein resource is rich in 8 kinds of essential amino acids, that is needed by the human body, and has widely used in food due to its high utilization rate and useful functional properties [2].

-          It is important to know some of these useful and functional properties of peanut protein.

 In recent years, many studies have focused on adding enzymes to the emulsion to change the emulsion’s stability.

-          Please insert some references of theses studies after the sentence.

 Line 178

The fatty acid compositions of peanut oil by various extract approaches were determined using gas chromatography (GC) by Gao et al [19].

-          Add some details regarding the identification process of fatty acids and databases used.

 Line 189

The AOCS official techniques (Cd 3d-63, Cd 8b-90, Cd 1c-85, and Cd 3-25, respectively) were used to calculate the acid value, peroxide value, Iodine value, and saponification value.

-          Please add a supplementary material section to give more details about the AOCS techniques used to determine the physicochemical properties of the oil.

 Line 275

Figure.2 The influence of different fatty acids on free oil yield.

-          You need to add the statistical significance of small letters used in the figure 2.

 Line 413

Table 5 Fatty acid composition of peanut oils obtained by different demulsification methods.

-          It is important to explain all abbreviation used in table 5, such as SE, HD, FTHD, IPD, SFA, UFA.

-          Also, it is necessary to explain the small letters and their statistical significance.

-          Same comment with table 6 and 7.

 Author Response

Dear reviewer,

We are appreciated for your comments and we have made improvement. Please see the attachment.

Dr. Chenxian Yang

Line 15

The analysis of the fatty acid composition and physicochemical characterization of oil extracted by four different methods was studied during AEE process

-          Please insert some important results regarding the chemical composition and physicochemical properties of oil.

  Reply: We are thankful for the suggestions. The results about chemical composition and physicochemical properties of oil were summarized again in abstract.

Line 17

Peanut oil extracted from demulsification with heptanoic acid contained more unsaturated fatty acids.

-          Please give more details about the quantity of unsaturated fatty acids obtained.

  Reply: We are thankful for your professional suggestions. The more details about the quantity of unsaturated fatty acids obtained were improved.

Line 18

And the oil extracted by heptanoic acid demulsification had 18 more acid value and lower peroxide values

-          Same comment as reported earlier.

 Reply: Firstly, we are thankful for the suggestions. Secondly, the issues mentioned above (line 15,17 and 18) is in abstract, we have re-summarized the results for oils and proteins in the abstract, which have been highlighted in the paper.

Lines 32-34

Moreover, peanut protein as a high-quality edible plant protein resource is rich in 8 kinds of essential amino acids, that is needed by the human body, and has widely used in food due to its high utilization rate and useful functional properties [2].

-          It is important to know some of these useful and functional properties of peanut protein.

 Reply: The authors are thankful for this suggestion. We also think that function properties of peanut proteins are important, we have simply added its importance in the manuscript.

In recent years, many studies have focused on adding enzymes to the emulsion to change the emulsion’s stability.

-          Please insert some references of these studies after the sentence.

Reply: The authors are thankful for this suggestion. We have inserted some references after the sentence.

Line 178

The fatty acid compositions of peanut oil by various extract approaches were determined using gas chromatography (GC) by Gao et al [19].

-          Add some details regarding the identification process of fatty acids and databases used.

 Reply: The authors are thankful for this suggestion. We have added the details about the identification process of fatty acid composition and databases used.

Line 189

The AOCS official techniques (Cd 3d-63, Cd 8b-90, Cd 1c-85, and Cd 3-25, respectively) were used to calculate the acid value, peroxide value, Iodine value, and saponification value.

-          Please add a supplementary material section to give more details about the AOCS techniques used to determine the physicochemical properties of the oil.

  Reply: The authors are thankful for this suggestion. We have added supplementary materials, and the detail about determination was shown in supplementary material S1.

Line 275

Figure.2 The influence of different fatty acids on free oil yield.

-          You need to add the statistical significance of small letters used in the figure 2.

 Reply: We are thankful for your valuable suggestion. The statistical significance of small letters used in the figure 2 were added.

Line 413

Table 5 Fatty acid composition of peanut oils obtained by different demulsification methods.

-          It is important to explain all abbreviation used in table 5, such as SE, HD, FTHD, IPD, SFA, UFA.

-          Also, it is necessary to explain the small letters and their statistical significance.

-          Same comment with table 6 and 7.

Reply: We are thankful for your valuable suggestion. We have improved the figure caption and explained all abbreviation in Table 5, Table 6 and Table 7. Also, the small letters and the statistical significance of all figures and tables in this paper were improved.